# Multi-Omics Analysis of Survival-Related Splicing Factors and Identifies CRNKL1 as a Therapeutic Target in Esophageal Cancer

**DOI:** 10.3390/genes16040379

**Published:** 2025-03-27

**Authors:** Tianrui Gao, Meiling Fan, Zhongyuan Zeng, Lixia Peng, Chao-Nan Qian, Xia Zhao, Bijun Huang

**Affiliations:** 1State Key Laboratory of Oncology in South China, Guangdong Key Laboratory of Nasopharyngeal Carcinoma Diagnosis and Therapy, Guangdong Provincial Clinical Research Center for Cancer, Sun Yat-sen University Cancer Center, Guangzhou 510060, China; gaotr@sysucc.org.cn (T.G.); fanml@sysucc.org.cn (M.F.); penglx@sysucc.org.cn (L.P.); 2Department of Experimental Research, Sun Yat-sen University Cancer Center, Guangzhou 510060, China; 3Lab Teaching & Management Center, Institute of Life Science and Laboratory of Tissue and Cell Biology, Chongqing Medical University, Chongqing 400016, China; zengzy2019220176@163.com; 4Department of Radiation Oncology, Guangzhou Concord Cancer Center, Guangzhou 510060, China; chaonan.qian@ccm.cn; 5Department of Microbiology, Army Medical University, Chongqing 400038, China

**Keywords:** alternative splicing, esophageal cancer, splicing factors

## Abstract

**Background:** RNA alternative splicing represents a pivotal regulatory mechanism of eukaryotic gene expression, wherein splicing factors (SFs) serve as key regulators. Aberrant SF expression drives oncogenic splice variant production, thereby promoting tumorigenesis and malignant progression. However, the biological functions and potential targets of SFs remain largely underexplored. **Methods:** Through multi-omics analysis, we identified survival-related splicing factors (SFs) in esophageal cancer and elucidated their biological regulatory networks. To further investigate their downstream splicing targets, we combined alternative splicing events resulting from SF knockdown with those specific to esophageal cancer. Finally, these splicing events were validated through full-length RNA sequencing and confirmed in cancer cells and clinical specimens. **Result:** We identified six SFs that are highly expressed in esophageal cancer and correlate with poor prognosis. Further analysis revealed that these factors are significantly associated with immune infiltration, cancer stemness, tumor heterogeneity, and drug resistance. *CRNKL1* was identified as a hub SFs. The target genes and pathways regulated by these SFs showed substantial overlap, suggesting their coordinated roles in promoting cancer stemness and metastasis. Specifically, alternative splicing of key markers, such as CD44 and *CTTN*, was regulated by most of these SFs and correlated with poor prognosis. **Conclusions:** Our study unveils six survival-related SFs that contribute to the aggressiveness of esophageal cancer and *CTTN* and *CD44* alternative splicing may act as common downstream effectors of survival-related SFs. This study provides mechanistic insights into SF-mediated tumorigenesis and highlight novel therapeutic vulnerabilities in esophageal cancer.

## 1. Introduction

Esophageal cancer is one of the most aggressive malignancies worldwide, characterized by high mortality rates and limited therapeutic options [1,2,3,4]. Despite advances in multimodal therapies, the 5-year survival rate for patients with advanced-stage disease remains dismal, underscoring the urgent need to identify novel molecular drivers and therapeutic targets [5]. Alternative splicing (AS), a pivotal post-transcriptional regulatory mechanism, generates transcriptomic diversity and is frequently dysregulated in cancer [6,7,8]. Splicing factors (SFs), the master regulators of AS, have emerged as critical players in tumorigenesis, metastasis, and therapy resistance [9,10]. However, their roles in esophageal cancer progression, particularly their functional networks and downstream effectors, remain poorly understood.

Recent studies have implicated SF dysregulation in tumor immune evasion, stemness maintenance, and drug resistance across multiple cancer types. For instance, factors such as *SF3B1* and *RBFOX2* have been extensively studied in cancer biology [11,12]. In esophageal cancer, however, systematic analyses of SFs are relatively scarce. Furthermore, while individual SFs have been studied, their downstream oncogenic splicing targets remain unexplored. There is a critical gap in understanding how SF-driven splicing networks converge on key pathways to drive malignant phenotypes in esophageal cancer, such as cancer stemness and immune suppression.

Here, we integrated multiple RNA-seq datasets and functional analyses to identify survival-related SFs in esophageal cancer and elucidate their biological networks. Our study systematically identifies six survival-associated SFs (*CRNKL1*, *SNRPB2*, *RBMX2*, *DDX46*, *PPWD1*, and *CFAP20*) that are functionally interconnected and correlate with poor prognosis. We explored their biological functions on immune infiltration, cancer stemness, and drug resistance. We further demonstrate that survival-related splicing factors co-regulate overlapping oncogenic pathways and identify *CTTN* and *CD44* splicing isoforms as key effectors of SF-driven malignancy. Notably, we reveal *CRNKL1* as a hub splicing factor regulating cytoskeletal dynamics and stemness. These findings provide insights into understanding SF-mediated oncogenesis and highlight novel therapeutic vulnerabilities in esophageal cancer.

## 2. Materials and Methods

### 2.1. Clinical Samples

Five paired esophageal cancer specimens and adjacent normal tissues were collected from Gaozhou People’s Hospital (Guangdong, China) for long-read sequencing analysis. An independent cohort comprising 10 paired ESCC samples was obtained from the First Affiliated Hospital of Sun Yat-Sen University (FAHSYSU) and Sun Yat-Sen University Cancer Center (SYSUCC). RNA was extracted using TRIzol reagent (Invitrogen, Carlsbad, CA, USA), aliquoted, and stored at −80 °C until analysis. Additionally, 54 formalin-fixed, paraffin-embedded ESCC tissues and matched adjacent normal specimens were acquired from SYSUCC. All procedures were approved by the Institutional Review Boards of FAHSYSU and SYSUCC, with written informed consent obtained from participants.

### 2.2. RNA-Sequencing

For the public database, we acquired raw RNA-seq data of SFs-knockdown from Malgorzata E. Rogalska et al. The detailed sequencing parameters are described in a previous publication [13]. We used rMATS-turbo v4.3.0 [14] to quantify AS changes. Percent spliced in (PSI) was used to measure AS events, and |△PSI| > 10% were regarded as indicative of differential AS events. Differential splicing events in esophageal cancer were obtained from Oncosplicing [15], which is based on The Cancer Genome Atlas (TCGA) database.

For our own sequencing data, we collected five paired ESCC specimens and adjacent normal tissues and performed ONT-nanopore full-length sequencing. A total of 1 ug of RNA was prepared for cDNA libraries using the cDNA-PCR Sequencing Kit (SQK-LSK110+EXP-PCB096) protocol provided by Oxford Nanopore Technologies (ONT). Briefly, the template switching activity of reverse transcriptases enriches for full-length cDNAs and adds defined PCR adapters directly to both ends of the first-strand cDNA. Following this, cDNA PCR for 14 cycles was performed using LongAmp Tag (NEB). The PCR products were then subjected to ONT adaptor ligation using T4 DNA ligase (NEB). Agencourt XP beads were used for DNA purification according to the ONT protocol. After filtering, mapping, and removing redundancies, PSI-sigma [16] was used to detect AS events. Events with |△PSI| > 10% in at least two paired clinical samples were regarded as differential AS events. KEGG analysis was performed using Sangerbox [17].

For scRNA-seq, we used scCancerExplorer v1.0 for analysis [18].

### 2.3. Cell Culture

Human esophageal cancer cell lines (KYSE30, KYSE150, KYSE180, KYSE410, and KYSE520) and immortalized esophageal epithelial cells (NE1) were obtained from the University of Hong Kong (Department of Clinical Oncology, The University of Hong Kong, Hong Kong, China. ). Eca109 cells were acquired from the BeNa Culture Collection (Beijing, China). Cells were maintained in DMEM (Gibco, Grand Island, NY, USA) supplemented with 10% fetal bovine serum (FBS; Gibco) and 1% penicillin–streptomycin (Sigma-Aldrich, St. Louis, MO, USA) at 37 °C under 5% CO_2_. Cell line authenticity was verified via STR profiling.

### 2.4. siRNA Synthesis

siRNA was purchased from Tsingke Biotech (Beijing, China), while primers were synthesized by Ruibio Biotechnology (Guangzhou, China). Plasmids were obtained from Umine Biotechnology (Guangzhou, China). The sequence of si*CRNKL1*-1 was 5′-GGGUACGAGUGAUUUAC-3′and si*CRNKL1*-2 was 5′-GACGUCGAUGAGAGUGA-3′.

### 2.5. Cell Transfection

For transient transfection, siRNAs (50 nM) were delivered using Lipofectamine 3000 (Thermo Fisher, Waltham, MA, USA) according to the manufacturer’s protocol.

### 2.6. RNA Isolation, RT-PCR, and RT-qPCR

Total RNA was isolated using the Universal RNA Purification Kit (EZB-RN4, EZBioscience, Minneapolis, MN, USA.). A total of 1 μg of RNA was reverse-transcribed using the Color Reverse Transcription Kit (EZBioscience, A0010CGQ). For RT-PCR analysis, cDNA served as the template for PCR amplification using Green Taq Mix (Vazyme, Nanjing, China P131-01). PCR products were separated on 2% agarose gel (Beyotime, Nantong, China), and images captured using the Geldoc Go system equipped with Image Lab Touch 4.0 software (Bio-Rad, Hercules, CA, USA). For RT-qPCR, cDNA was used as the template for qPCR, performed with 2× Color SYBR Green qPCR Master Mix (EZBioscience, A0012-R2) on a LightCycler 480 system (Roche, Basel, Switzerland). The sequence of *CTTN* primers was as follows: Forward: 5′-AAAGTGGATAAGAGCGCCGT-3′, Reverse: 5′-CATACTTCCCGCCGAATCCT-3′. The sequence of *CD44s* primers was Forward: 5′-GCTTCAGCCTACTGCAAATCC-3′, Reverse: 5′-CGTTGTCATTGAAAGAGGTCCTG-3′. The sequence of *CD44v8* primers was Forward: 5′-AAGACATCTACCCCAGCAAC-3′, Reverse: 5′-TCTCTGGTAGCAGGGATTC-3′. The sequence of *GAPDH* primers was Forward: 5′-GGAGCGAGATCCCTCCAAAAT-3′, Reverse: 5′-GGCTGTTGTCATACTTCTCATGG-3′.

### 2.7. Western Blotting

Cells were lysed with RIPA lysis buffer (Beyotime, Nantong, China), supplemented with protease and phosphatase inhibitors. Protein concentrations were measured using the Pierce™ BCA Protein Assay (Thermo Fisher Scientific). Cell lysates were separated on SDS-PAGE gels and transferred to PVDF membranes (Merck Millipore, Darmstadt, Germany). Hsp70 primary antibody (PB9638, Boster, Pleasanton, CA, USA) and CRNKL1 primary antibody (2212C1a, SANTACRUZ) were used for Western blotting.

### 2.8. Immunofluorescence

A total of 1 × 10^5^ cells were cultured in confocal dishes (Nest Biotechnology, Ningbo, China) until they reached a density of 50–70%. The cells were then harvested and fixed with 4% paraformaldehyde. After fixation, the cells were incubated overnight with a primary antibody, followed by incubation with a secondary antibody for one hour. Cell nuclei were stained using DAPI (Beyotime). Images were captured and visualized using a Zeiss LSM800 Meta laser scanning confocal microscope (Carl Zeiss, Jena, Germany).

### 2.9. Immunohistochemistry (IHC)

The immunohistochemical assay procedures followed those described by other researchers [19]. The SNRPB2 antibody (13512-1-AP, Proteintech, Wuhan, China) was used for IHC.

### 2.10. Cell Proliferation Assays (CCK-8)

1 × 10^3^ ESCC cells per well were cultured in 96-well cell culture dishes in six replicates. After the cells adhered to the dishes, the relative cell number was measured using the Cell Counting Kit 8 (CCK-8) (TargetMol, Shanghai, China) according to the manufacturer’s instructions every day. After seven days, growth curves were plotted based on 405 nm OD values.

### 2.11. Colony Formation Experiment

A total of 2 × 10^3^ ESCC cells per well were cultured in 6-well cell culture dishes. After 14 days, the cells were fixed with methanol and stained with 1% crystal violet. The stained cells were then counted for statistical analysis.

### 2.12. Sphere Formation Experiment

1 × 10^3^ ESCC cells per well were cultured in 6-well cell culture dishes. After 10–14 days, tumor spheres could be observed under a microscope. The number of tumor spheres was then counted for statistical analysis.

### 2.13. In Vitro Migration and Invasion Assays

Transwell assays were performed to assess in vitro cell migration and invasion. For migration, 1–5 × 10^5^ ESCC cells were seeded into the upper chambers of the transwell inserts. For invasion assays, the upper chambers were pre-coated with Matrigel (Corning, Corning, NY, USA) overnight before seeding 1–5 × 10^5^ ESCC cells, following the same protocol as for migration. After 20 h, the cells that migrated into or invaded the lower chamber were fixed with methanol and stained with 1% crystal violet. The stained cells were then counted under a microscope.

### 2.14. Wound Healing Assays

A total of 2 × 10⁵ cells per well were seeded into six-well cell culture dishes. Sterile tips were used to create scratch wounds in the cell monolayers. The width of the wounds was captured and measured using microscopy at designated time points. The wound healing speed was normalized by calculating the migration index, using the following formula:Migration index = (1 − (scratch area of 12/24/36/48 h)/(scratch area at 0 h)) %

### 2.15. Statistical Analysis

Publicly accessible bioinformatics tools were utilized as follows: UALCAN (cancer biomarker discovery), GEPIA2 (gene expression profiling), Cytoscape 3.10.0 (network visualization), Sangerbox (multi-omics analysis), OncoSplicing (alternative splicing analysis), and scCancerExplorer (single-cell RNA-seq analysis). Statistical analyses were automatically performed using each platform’s default parameters. For our own data analysis, statistical analyses were performed using SPSS 26.0 (IBM, Armonk, NY, USA). Continuous variables are expressed as mean ± standard deviation (SD). Inter-group comparisons were conducted based on data distribution and sample characteristics: Normally distributed data between two groups were compared using the independent Student’s *t*-test; non-normally distributed data were analyzed with the Mann–Whitney U test. A two-tailed *p* < 0.05 was considered statistically significant. Graphical representations of experimental results were created using GraphPad Prism 8.0 (GraphPad Software, San Diego, CA, USA)

## 3. Results

### 3.1. Identification of Survival-Related SFs in Esophageal Cancer

To identify SFs correlated with poor prognosis in esophageal cancer, we summarized the prognostic markers from GEPIA2 and UALCAN, based on the TCGA database. A total of 1406 prognostic markers were identified in esophageal cancer, and we then screened SFs among these markers. Ultimately, six survival-related SFs (*CRNKL1*, *SNRPB2*, *RBMX2*, *DDX46*, *PPWD1*, and *CFAP20*) were selected. The detailed workflow is shown in Figure 1.

In RNA-seq data from TCGA and proteomics data [20] from ESCC, the six SFs listed above were both elevated and correlated with poor prognosis (Figure 2A–C). We selected two of these SFs (SNRPB2, CRNKL1) and validated their aberrant expression in paired esophageal cancer tissues (Figure 2D,E). We used the STRING database to explore the protein–protein interaction network between the survival-related SFs. Interestingly, five of the six survival-related SFs were functionally correlated. GO enrichment analysis showed that these five SFs may be involved in U2-type spliceosome assembly and mRNA splicing (Figure 2F). Furthermore, the mRNA expression levels of these survival-related SFs were also significantly correlated in the TCGA database (Figure 2G). These results indicate that survival-related SFs in esophageal cancer are closely connected in terms of biological function and may co-regulate esophageal cancer malignant activity.

### 3.2. Survival-Related SFs Correlate with Immune Infiltration, Cancer Stemness, and Drug Resistance in Esophageal Cancer

To further explore the biological functions of survival-related SFs, we performed a comprehensive bioinformatics analysis. Immune evasion and the tumor microenvironment (TME) are recognized as important components in tumor biology and cancer treatment. We used the EPIC algorithm [21] to estimate the proportions of tumor-infiltrating immune cells. Almost all survival-related SFs were negatively correlated with CD8^+^ cells (Figure 3A). *CFAP20, PPWD1,* and *DDX46* were positively correlated with cancer-associated fibroblasts (CAFs) (Figure 3B). To explore the expression of survival-related SFs in tumor microenvironment, we performed single-cell RNA sequencing analysis (scRNA-seq). Results revealed that most survival-related splicing factors (SFs) were predominantly expressed in cancer cells compared to other immune cells (Figure 3C), suggesting that these SFs may primarily influence the tumor microenvironment by acting within the cancer cells themselves. All survival-related SFs expression in ESCA scRNA-seq were summarized in Appendix A. We performed immunohistochemistry (IHC) of SNRPB2 and tumor-infiltrating CD8^+^ T cells in esophageal cancer for validation. SNRPB2 expression was negatively correlated with levels of tumor-infiltrating CD8^+^ T cells (Figure 3D), suggesting that survival-related SFs may be involved in tumor immune escape and microenvironment changes.

Next, we explored the relationship between survival-related SFs and immune modulators. The results showed that survival-related SFs significantly correlated with several chemokines, immune stimulators, and inhibitors. We found that *CCL15*, a chemokine known to promote an immunosuppressive microenvironment and cancer metastasis, was significantly correlated with *CRNKL1*, *SNRPB2*, and *PPWD1* (Figure 3E). This suggests that these splicing factors (SFs) may play a key role in modulating the tumor microenvironment (TME). The relationship between survival-related SFs and immune modulators is summarized in Appendix A. Additionally, several survival-related SFs (*CRNKL1* and *DDX46*) were positively correlated with tumor purity (Figure 3F), suggesting lower tumor immune infiltration.

Cancer stemness is a crucial characteristic of cancer cells, contributing to immune escape, drug resistance, and metastasis. We analyzed the relationship between survival-related SFs and cancer stemness using the RNA based Stemness Scores (RNAss) [22]. *CRNKL1* and *SNRPB2* showed positive correlations with esophageal cancer stemness (Figure 4A), suggesting that survival-related SFs may enhance esophageal cancer stemness. Tumor heterogeneity is highly correlated with cancer stemness. Most survival-related SFs significantly correlated with tumor heterogeneity (Figure 4B), as calculated by the mutant-allele tumor heterogeneity (MATH) score [23]. Aneuploidy, driven by multiple types of genetic alterations, is vital for promoting tumorigenesis and tumor evolution by enhancing tumor heterogeneity [24]. We explored the relationship between survival-related SFs and ploidy, and found that *CRNKL1, SNRPB2, RBMX2*, and *CFAP20* showed significant positive correlations with tumor ploidy (Figure 4C). These findings suggest that survival-related SFs are also related to tumor heterogeneity.

Finally, we explored the relationship between survival-related SFs and drug resistance using the CAPDS database [25]. In brief, most survival-related SFs were associated with resistance to chemotherapy drugs or targeted drugs. For instance, *CRNKL1* was linked to oxaliplatin, cisplatin, and 5-fluorouracil resistance (Figure 4D). All survival-related SFs, except *CFAP20*, were associated with resistance to tyrosine kinase inhibitors (Figure 4E). Notably, several pathway inhibitors, including Wnt-C59 and JNK Inhibitor VIII, showed greater resistance in *CRNKL1* and *SNRPB2* overexpressing groups (Appendix A). All drug resistance data of survival-related SFs were summarized in Appendix A.

The evidence presented above suggests that survival-related SFs in esophageal cancer may affect immune escape, cancer stemness, and drug resistance, contributing to poor prognosis.

### 3.3. Identification of CRNKL1 as a Prognostic Marker for Esophageal Cancer

To identify the critical splicing factor in esophageal cancer, we used Cytoscape [26] to identify hub genes among the survival-related SFs. *CRNKL1* was ranked first among the six survival-related SFs (Figure 5A). Additionally, the *CRNKL1* had the highest number of splicing targets and potential resistant drugs among the six survival-related SFs.

We find that *CRNKL1* was correlated with esophageal cancer tumor grade (Figure 5B). As mentioned above, *CRNKL1* was correlated with low immune infiltration and drug resistance. KEGG analysis revealed that the splicing targets modulated by *CRNKL1* are primarily involved in tight junctions, adherens junctions, and proteoglycans in cancer (Figure 6C), suggesting that *CRNKL1* may be linked to migration, invasion, and metastasis.

We performed in vitro functional experiments to confirm whether *CRNKL1* could affect the phenotype of esophageal cancer cells. Knockdown of *CRNKL1* significantly impaired the migration and invasion abilities of KYSE30 and KYSE410 cells (Figure 5C–G). Phalloidin staining was used to visualize the cytoskeleton. As expected, manipulation of *CRNKL1* expression significantly remodeled the cytoskeleton of both KYSE30 cells (Figure 5H). Sphere formation experiments showed that knockdown of *CRNKL1* significantly impaired cancer stemness in KYSE30 and KYSE410 cells (Figure 5I). Colony formation assays and cell proliferation growth curves indicated that *CRNKL1* can promote esophageal cancer proliferation (Figure 5J,K).

These results suggest that *CRNKL1* may function as a prognostic marker that influences cancer metastasis and stemness in esophageal cancer.

### 3.4. The Downstream Splicing Targets Regulated by Survival-Related SFs Are Highly Similar

To further explore the mechanism by which survival-related SFs promote esophageal cancer malignancy, we analyzed the AS changes following the knockdown of survival-related SFs from Malgorzata E. Rogalska et al. [13]. After intersecting SFs-specific splicing targets with esophageal-specific splicing events from Oncosplicing [15], we identified individual survival-related SF-driven AS targets in esophageal cancer (Figure 6A). In total, 2266 AS events were screened, with exon skipping events (ES) accounting for 77.18% of all splicing events (Figure 6B). The number of splicing targets among survival-related SFs showed minimal differences, with *CRNKL1* and *SNRPB2* having the most splicing targets.

Interestingly, KEGG analysis revealed that the splicing targets of the six survival-related SFs were partially similar, primarily focusing on cellular community-related pathways (focal adhesion, tight junction, adherens junction, ECM-receptor interaction), regulation of the actin cytoskeleton, and stem cell signaling pathways (Wnt signaling pathway, Hedgehog signaling pathway) (Figure 6C). All of these pathways are recognized as playing a significant role in metastasis, drug sensitivity, and tumorigenesis. We then screened for common splicing targets and performed KEGG enrichment analysis, which highlighted the Wnt and Hedgehog signaling pathways, tight junctions, ECM-receptor interaction, and chemokine signaling pathway as enriched.

These results suggest that survival-related SFs may co-regulate common oncogenic pathways in esophageal cancer to promote its progression.

### 3.5. Oncogenic AS Events, Including Cortactin and CD44, Act as Common Downstream Effectors of Survival-Related SFs

To identify key splicing targets regulated by survival-related SFs, we searched among the common splicing targets identified above. Considering that survival-related SFs are associated with stem cell signaling pathways and cytoskeleton remodeling, we selected cortactin (*CTTN*) and *CD44* as candidates (Figure 7A). Cortactin, a member of actin-binding protein family, plays an important role in cancer cell migration, cytoskeleton remodeling, and Epithelial–Mesenchymal Transition (EMT) [27]. Cortactin has been reported to have three isoforms: WT (with both exon 10 and exon 11 included), SV1 (with exon 11 skipped) and SV2 (with both exon 10 and exon 11 skipped) [28] (Figure 7B).

The *CTTN* exon 11 skipping/inclusion event was observed both in esophageal cancer samples and survival-related SFs knockdown data. Exon 11 skipping of *CTTN* occurred after knockdown of most survival-related SFs (*CRNKL1*, *SNRPB2*, *PPWD1*, *RBMX2*, and *DDX46*), while exon 11 inclusion was observed in esophageal cancer in comparison to normal tissue, both in the TCGA database and in our own full-length sequencing data (Figure 7C–E). This suggests that *CTTN* exon 11 inclusion may be a key event in esophageal cancer development and is controlled by the SFs mentioned above. To explore whether *CTTN* promotes ESCA metastasis by alternative splicing rather than mRNA overexpression, we analyzed *CTTN* expression and gene dosage in public ESCA database (Appendix A). We did not observe a significant amplification of *CTTN* in ESCA, and no significant differences in *CTTN* mRNA levels between ESCA and normal tissues. This suggests *CTTN* may promote ESCA malignant behaviors mainly by alternative splicing rather than by overexpression.

We then validated *CTTN* exon 11 inclusion in paired esophageal cancer samples. AlphaFold predicted an interaction between *CTTN* exon 11 RNA and *CRNKL1* (Figure 7F). Exon 11 skipping was observed following the knockdown of *CRNKL1* in KYSE30 and KYSE410 cells (Figure 7G–H). Interestingly, exon 11 inclusion was observed in both esophageal cancer cell lines and clinical specimens (Figure 7I–J). Additionally, *CTTN* exon 11 inclusion was correlated with poor prognosis in esophageal cancer (Figure 7K). These results suggest that *CTTN* exon 11 inclusion, co-regulated by survival-related SFs, is a prognostic marker for esophageal cancer.

We also found that CD44 AS is a common splicing target in esophageal cancer. Similar to *CTTN, CD44* splicing was observed in clinical samples and was correlated with poor prognosis (Figure 7Q). As a cell-surface glycoprotein, *CD44* is recognized as an important marker of cancer stemness, metastasis, and ECM interactions [29]. *CD44* AS has been reported to be relevant to metastasis and EMT in many types of cancer. *CD44* has multiple isoforms due to AS, and we observed *CD44* exon 12 (*CD44v8*) skipping in esophageal cancer, resulting in a change in the proportion between *CD44v8-10* and *CD44s* (the standard isoform) (Figure 7L–O). We validated that the expression of *CD44v8* decreased, while *CD44s* increased after knockdown of *CRNKL1* in KYSE30 cells (Figure 7P).

The evidence presented above reveals that several oncogenic isoforms and pathways are co-regulated by upstream survival-related SFs, contributing to metastasis and cancer stemness.

### 3.6. The Expression of Survival-Related Splicing Factors in Pan-Cancer

Finally, we explored the expression of survival-related SFs across multiple cancers using data from the TCGA and CPTAC databases. Overall, we found that survival-related SFs were aberrantly elevated in most cancer types at both the mRNA and protein levels (Appendix A). This suggests survival-related splicing factors may act as common oncogenes in cancers.

## 4. Discussion

This study uncovers a group of survival-related SFs that may regulate common pathways and gene splicing to promote immune evasion, cancer stemness, cytoskeletal remodeling, and poor prognosis in esophageal cancer. By integrating bioinformatics, experimental validation, and clinical correlation analyses, we identified six SFs (*CRNKL1*, *SNRPB2*, *RBMX2*, *DDX46*, *PPWD1*, and *CFAP20*) that are aberrantly overexpressed in esophageal cancer and tightly linked to poor prognosis. These SFs exhibit functional synergy, as demonstrated by their co-regulated splicing programs that target key pathways, such as Wnt signaling, Hedgehog signaling, and tight junctions. These findings provide new insights into the biological functions and potential oncogenic mechanisms of SFs in esophageal cancer.

Notably, *CRNKL1* was identified as a hub splicing factor with previously unappreciated roles in endothelial cells. *CRNKL1*, a member of the crooked-neck family, demonstrates a high level of evolutionary conservation across various species, including *Homo sapiens*, *Mus musculus* and *Drosophila melanogaster* [30]. Situated within the cell nucleus, *CRNKL1* plays a pivotal role in the splicing of pre-mRNA and is an integral component of the spliceosome complex [30,31,32]. Recent studies have demonstrated that *CRNKL1* was significantly overexpressed in bladder carcinoma and potentially interacted with key tumor-driving gene mutations, particularly in *TP53*. Furthermore, emerging evidence suggests that *CRNKL1* might promote bladder tumorigenesis through dysregulation of cell cycle progression pathways, thereby contributing to malignant transformation [33]. Although *CRNKL1* is known to be involved in spliceosome assembly [32], its biological role remains unclear. Our results reveal for the first time that *CRNKL1* depletion impaired migration, stemness, and cytoskeletal organization in esophageal cancer cells. Additionally, *CRNKL1* regulates cortactin and *CD44* oncogenic AS, positioning it as a key factor linking splicing regulation to metastatic competence. Its association with reduced CD8^+^ T cell infiltration and drug resistance further suggests multifunctional roles in esophageal cancer. Our findings position *CRNKL1* as a prognostic marker and a potential target for immunotherapy in esophageal cancer.

Another key finding is the convergent regulation of key splicing events by multiple SFs. Specifically, the inclusion of *CTTN* exon 11 and CD44 exon 12 emerged as pivotal nodes in SF-mediated oncogenesis. *CTTN* exon 11 inclusion was detected both in the TCGA database and our cohort, and it was co-regulated by survival-related SFs, which may contribute to EMT and cancer stemness. Existing studies have shown that the overexpression of the WT isoform significantly enhances cellular motility, whereas overexpression of SV1 or SV2 leads to a markedly reduced ability to promote cell movement compared to the WT isoform [28,34]. Specifically, SV1 retains a pro-motility function but is less effective than the control group, whereas SV2 exhibits an inhibitory effect on cell movement. Mechanistically, SV2 has a significant weaker actin-binding affinity than WT; although SV1 has similar actin-binding affinity to WT, its actin cross-linking efficiency is significantly weaker [34]. This regulatory mechanism of alternative splicing has been well explored in nasopharyngeal carcinoma [28]. Specifically, *PTBP2* and *TIA1* have been identified as key regulators mediating *CTTN* pre-mRNA splicing, leading to an increased production of the *WT-CTTN* isoform. Functional studies have demonstrated that this splicing shift significantly enhances pseudopodium formation through *WT-CTTN* accumulation, which subsequently promotes tumor cell motility and drives metastatic progression. These findings collectively suggest that tumor cells exploit *CTTN* alternative splicing as a molecular strategy to augment their invasive capacity, primarily by elevating the relative abundance of the *WT-CTTN* splice variant [28,34]. Similarly, the AS of *CD44*, driven by SFs, may enhance stemness by reducing hyaluronan binding and potentiating Wnt/β-catenin signaling [35]. These findings align with studies in triple negative breast cancers and lung cancers, where *CD44* splicing variants dictate stem-like properties [36,37]. Our data highlight the biological importance of survival-related SFs as upstream regulators of oncogenic isoform switching, providing a new perspective on the molecular mechanisms of esophageal tumorigenesis.

The clinical implications of our study are twofold. First, the survival-related SFs and their regulated isoforms (e.g., *CTTN* exon 11, *CD44v8*) may serve as prognostic biomarkers or therapeutic targets. Small molecules targeting SFs, such as spliceostatin analogs, could disrupt oncogenic splicing programs. Antisense oligonucleotides targeting specific gene AS can also be a potential treatment for esophageal cancer. Second, the link between SFs and immune-cold TMEs suggests that SF inhibition might synergize with immunotherapy.

However, some limitations exist. Although our findings indicated that *CRNKL1* regulates the alternative splicing of *CTTN* and *CD44*, the underlying molecular mechanisms remain elusive and warrant further investigation. Our functional validations focused on *CRNKL1*, warranting further investigation into the roles of other SFs. In vivo experiments and molecular interaction studies, such as CLIP-seq, are needed to confirm the biological roles of survival-related SFs. Whether survival-related SFs work in a coordinated manner remains to be further explored. Moreover, the precise mechanisms by which SFs coordinately regulate shared targets remain to be elucidated.

## 5. Conclusions

In conclusion, this study unveils a network of survival-related SFs that orchestrate splicing programs that contribute to the aggressiveness of esophageal cancer. These SFs play a central role in cancer stemness, cytoskeleton remodeling, and immune evasion. Several downstream oncogenic isoforms regulated by survival-related SFs could serve as prognostic markers for esophageal cancer. Our findings provide valuable insights into potential new therapies based on SFs or specific isoforms in esophageal cancer.

## Figures and Tables

**Figure 1 genes-16-00379-f001:**
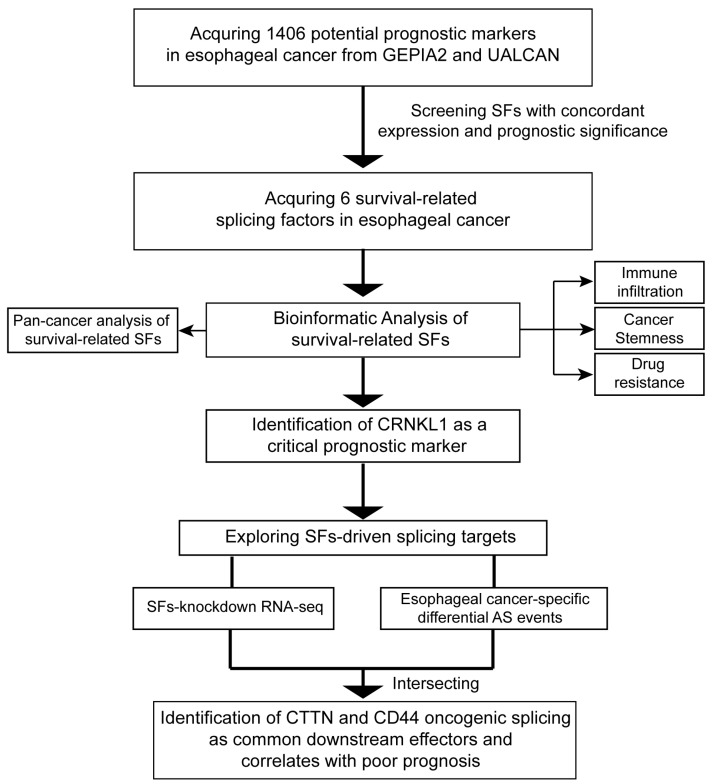
Workflow of data acquisition and analysis in this study.

**Figure 2 genes-16-00379-f002:**
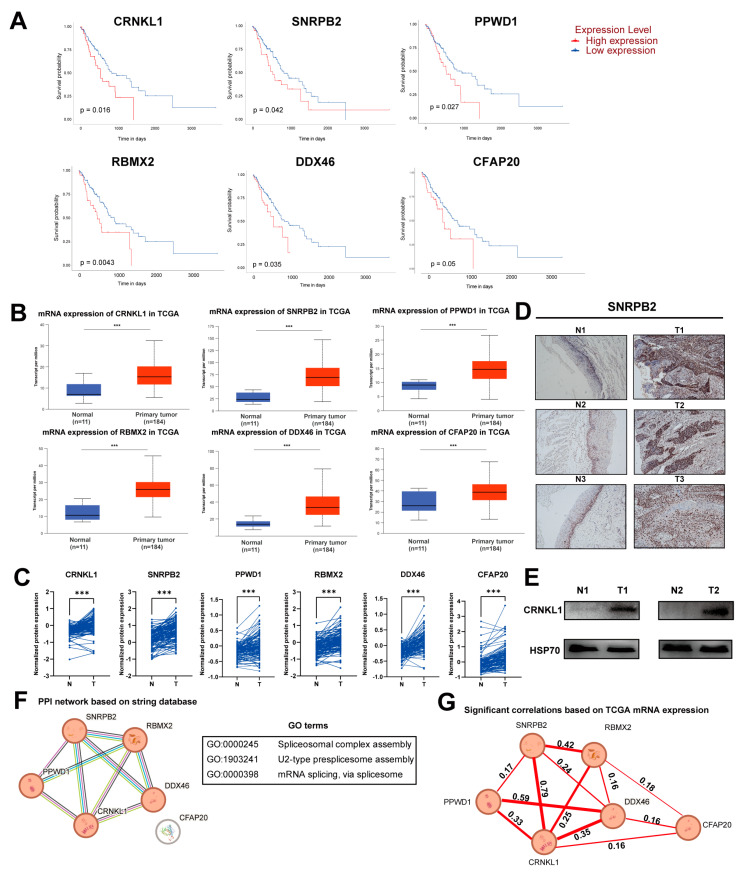
(**A**) Kaplan–Meier plot showing the survival analysis of six survival-related splicing factors (SFs) in esophageal cancer, based on TCGA database. (**B**) mRNA expression level of six survival-related SFs in normal and esophageal cancer tissues based on TCGA database. (**C**) The abnormal expression of survival-related SFs in esophageal cancer based on proteomics data. (**D**,**E**) Aberrant expression of *SNRPB2* and *CRNKL1* in esophageal cancer tissues. (**F**) Protein–protein interaction (PPI) network analysis using the STRING database demonstrated that five of the six survival-related SFs were functionally correlated. Gene Ontology (GO) enrichment analysis indicated that these SFs are involved in U2-type spliceosome assembly and mRNA splicing. (**G**) Analysis of mRNA expression levels of the survival-related SFs in the TCGA database showed significant correlations among the SFs. The thickness of lines represents the strength of correlation between SFs, with the correlation coefficient indicated along the lines. *** *p* < 0.001.

**Figure 3 genes-16-00379-f003:**
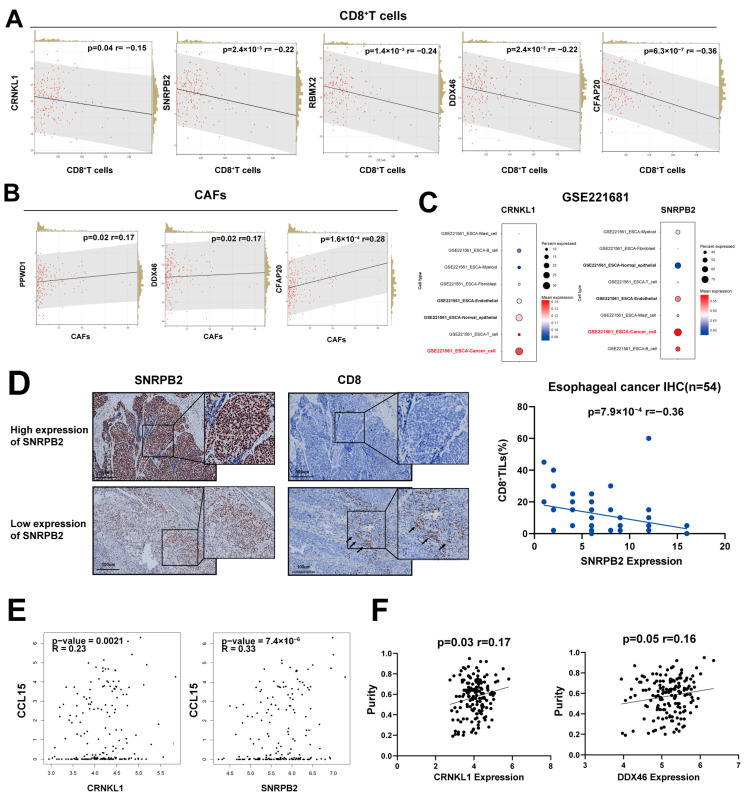
(**A**) Negative correlation between survival-related splicing factors (SFs) and levels of tumor-infiltrating CD8^+^ T cells. (**B**) Positive correlation between survival-related SFs and cancer-associated fibroblasts (CAFs). (**C**) Aberrant expression of survival-related splicing factors (SFs) in cancer cells shown by single-cell RNA sequencing. (**D**) Negative correlation between SNRPB2 expression and tumor-infiltrating CD8^+^ T cells, based on immunohistochemical (IHC) staining. Arrows indicate tumor-infiltrating CD8^+^ T cells. (**E**) Positive correlation between survival-related SFs and representative chemokines. (**F**) Positive correlation between survival-related SFs and tumor purity.

**Figure 4 genes-16-00379-f004:**
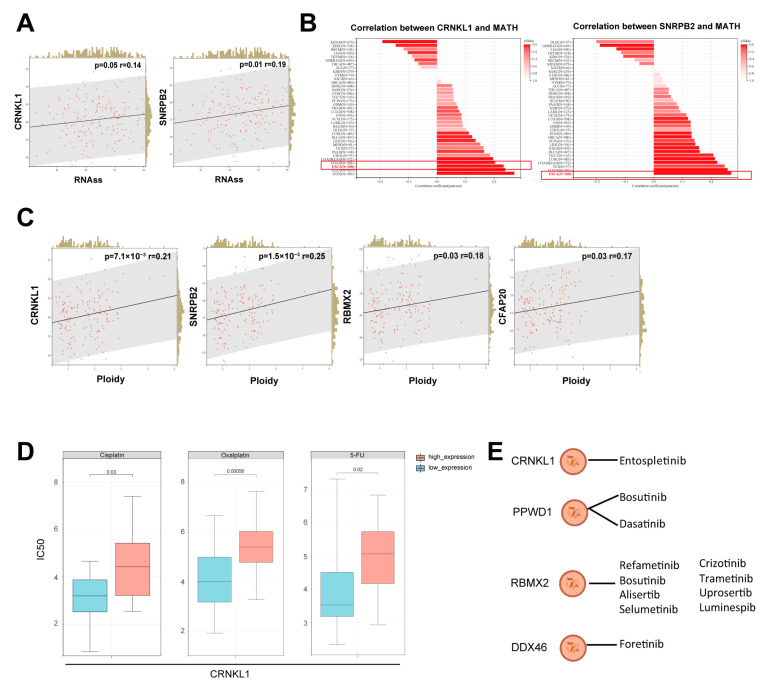
(**A**) Positive correlation between survival-related SFs and cancer stemness, as assessed by RNAss estimation. (**B**,**C**) Positive correlations between survival-related SFs and MATH (**B**) and ploidy (**C**). (**D**) Representative splicing factor *CRNKL1* is correlated with resistance to cisplatin, oxaliplatin, and 5-fluorouracil in esophageal cancer. (**E**) Survival-related SFs are associated with resistance to tyrosine kinase inhibitors.

**Figure 5 genes-16-00379-f005:**
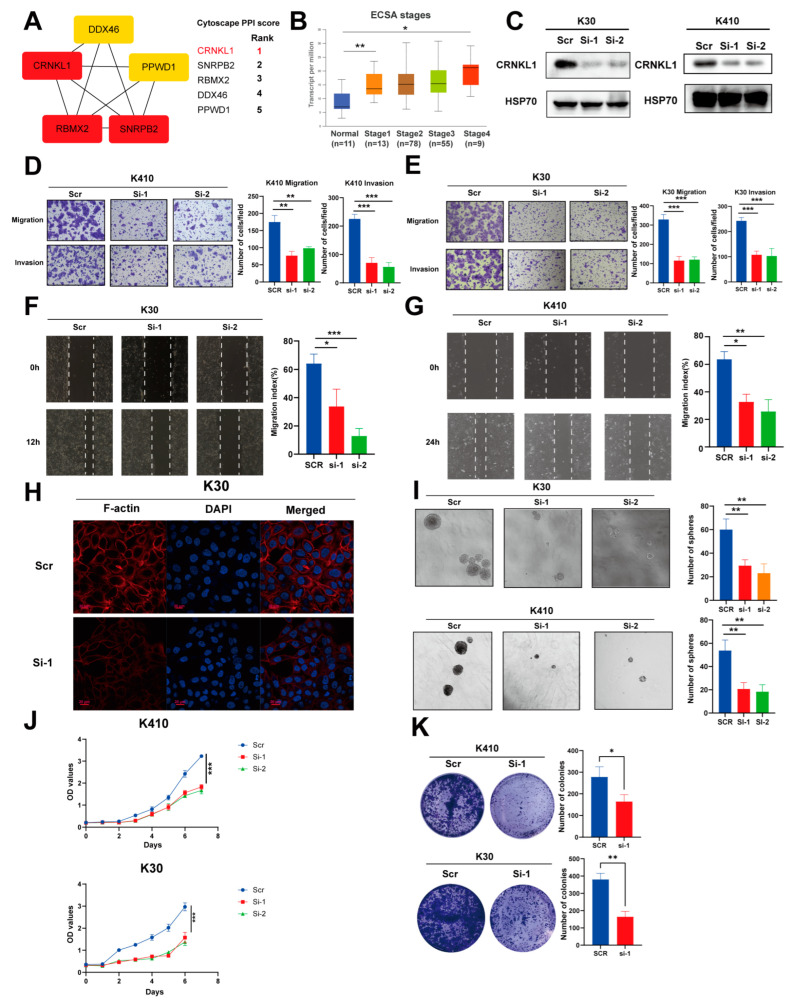
(**A**) Identification of hub survival-related splicing factors (SFs) using Cytoscape PPI score. (**B**) Correlation between *CRNKL1* expression and clinical stages of esophageal cancer (ESCA). (**C**) Validation of *CRNKL1* knockdown in KYSE30 and KYSE410 cells by Western blotting. (**D**–**G**) *CRNKL1* knockdown significantly impairs cell migration in KYSE30 and KYSE410 cells, as demonstrated by transwell assays (**D**,**E**) and wound healing assays (**F**,**G**). (**H**) *CRNKL1* knockdown significantly affects cytoskeleton remodeling in KYSE30 cells. (**I**) *CRNKL1* knockdown significantly impairs cancer stemness in KYSE30 and KYSE410 cells, as demonstrated by tumor sphere formation experiments. (**J**,**K**) *CRNKL1* knockdown significantly affects tumor proliferative ability, as indicated by proliferation curves (**J**) and colony formation assays (**K**). * *p* < 0.05, ** *p* < 0.01, *** *p* < 0.001.

**Figure 6 genes-16-00379-f006:**
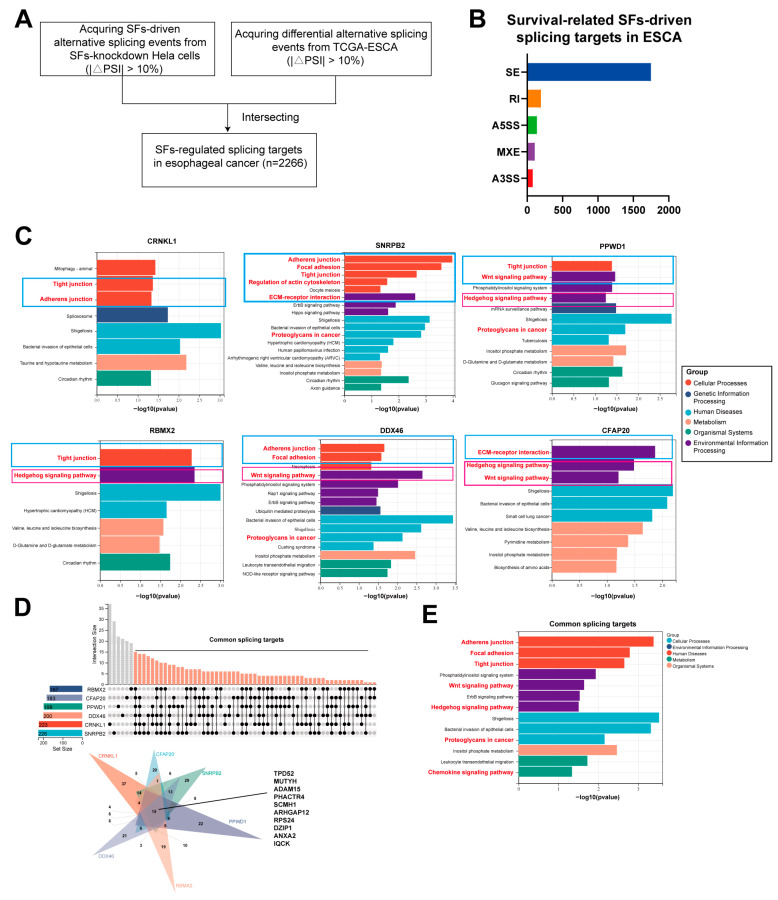
(**A**) Workflow for deciphering SFs-driven esophageal cancer-specific splicing targets. (**B**) Statistics of survival-related SFs-driven splicing targets in ESCA. (**C**) KEGG enrichment analysis of pathways enriched in SF-driven splicing targets. Blue and red rectangles showed cytoskeletal pathways and stemness cell pathways. (**D**) Common splicing targets driven by survival-related SFs in esophageal cancer. (**E**) KEGG enrichment analysis of common splicing targets.

**Figure 7 genes-16-00379-f007:**
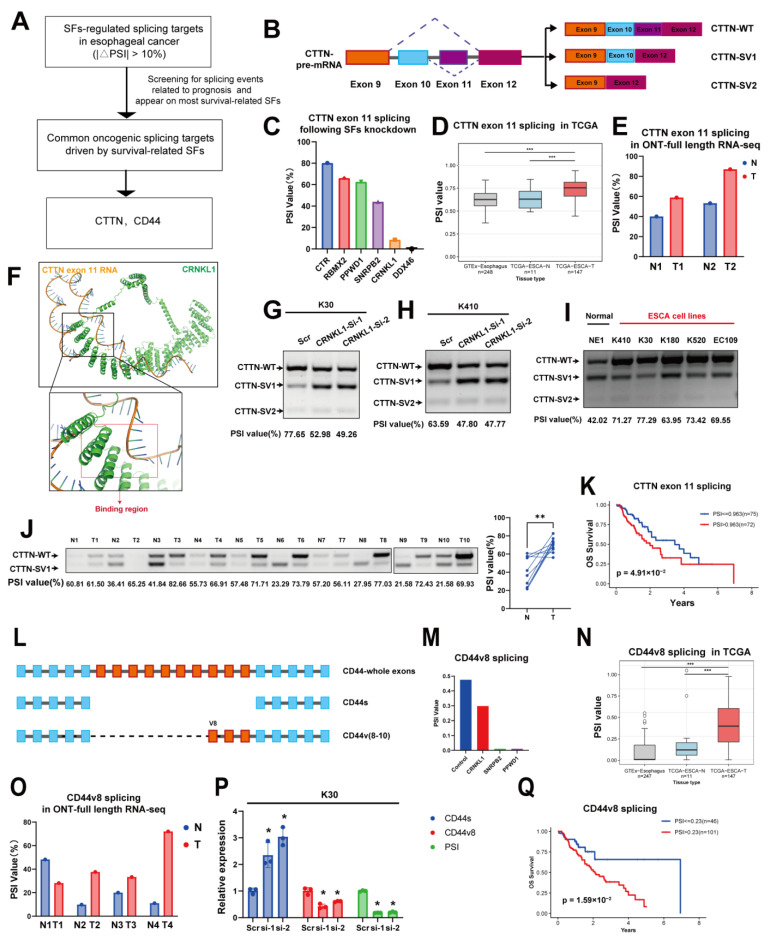
(**A**) Workflow for exploring common downstream splicing targets in ESCA. (**B**) Alternative splicing (AS) of *CTTN* exon 11 and its different isoforms. (**C**–**E**) Changes in *CTTN* exon 11 AS in SFs-knockdown HeLa cells (**C**), the TCGA database (**D**), and full-length RNA-seq of paired ESCA tissues (**E**). (**F**) AlphaFold3-predicted interaction between *CRNKL1* and *CTTN* exon 11 RNA. (**G**–**J**) Knockdown of *CRNKL1* significantly promotes *CTTN* exon 11 skipping in KYSE30 (**G**) and KYSE410 (**H**) cells; conversely, *CTTN* exon 11 inclusion was observed in both ESCA cell lines (**I**) and clinical specimens (**J**). (**K**) *CTTN* exon 11 inclusion is significantly correlated with poor prognosis in ESCA. (**L**) Illustration of different *CD44* isoforms. (**M**–**O**) Changes in *CD44v8* AS in SFs-knockdown HeLa cells (**M**), the TCGA database (**N**), and full-length RNA-seq of paired ESCA tissues (**O**). (**P**) Knockdown of *CRNKL1* significantly promotes *CTTN* exon 11 skipping in KYSE30 cells. (**Q**) *CD44v8* inclusion significantly correlated with poor prognosis in ESCA. * *p* < 0.05, ** *p* < 0.01, *** *p* < 0.001.

## Data Availability

Raw data of this study have been deposited to the Research Data Deposit database (www.researchdata.org.cn, accessed on 20 March 2025) under accession number RDDB2025453602.

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
