# Peer review of "Multi-Omics Analysis of Survival-Related Splicing Factors and Identifies CRNKL1 as a Therapeutic Target in Esophageal Cancer"

_genes, 2025, doi:10.3390/genes16040379_

Round 1

Reviewer 1 Report

Comments and Suggestions for Authors

The manuscript “Multi-omics analysis of survival-related Splicing Factors and 2 Identifies CRNKL1 as a Therapeutic Target in Esophageal Caner”  by Gao et al. describes the identification and characterization of six splicing factors overexpressed in esophageal cancer. The authors demonstrate that knocking down some of these genes reduced cancer cell proliferation and migration. They further identified alternative splicing events affecting several malignancy-associated genes. The study presents a clear and sound message. I only have a few minor concerns regarding data integrity and interpretation.

1.Several figures contain inconsistencies that should be addressed:

  • Figure 3F: The two scatter plots appear identical.
  • Figure 3D: The number of sampling dots (~30) is smaller than the indicated sample size (54).
  • Figures 7K and 7Q: The PSI cutoffs seem arbitrary and do not match the distributions in Figures 7D and 7N. For instance, CTTN PSI at 0.963 is above 1.5×IQR on the box plot, yet the sample sizes for the above and below groups are roughly equal.
  • Figure 4E: Incorrectly referred as 4F or 4H in the text.
  • Figure 5: The authors should specify sample size (N) and p-value thresholds.

2.The manuscript reports PPWD1 overexpression, but my access to GEPIA suggested it is downregulated in most cancer types including ESCA. Could the authors clarify this discrepancy?

3.A key observation is that overexpression of splicing factors in tumors enhances the proper splicing of CTTN into its "wild-type" isoform. However, the functional consequences of this shift remain unclear. Some discussions would be welcomed, for example whether this shift toward wild-type isoforms represents a pseudo-overexpression effect rather than an alternative oncogenic splicing event, or if such splicing-driven isoform balance affects functional gene dosage in TCGA and other databases.

Reviewer 2 Report

Comments and Suggestions for Authors

Major concerns and comments:

  1. For clinical samples, the sample size is 5. Please discuss the small sample size.
  2. Please add more info in the introduction of CRNKL1.
  3. Does CKNKL1 gene dosage increase in patients with esophageal cancers?
  4. Any database info supports the current study finding?
  5. In 2.3 Cell Culture session: Five cell lines were listed but only 2 cell lines were tested in the study. Any info from the other 3 cell lines (KYSE150, KYSE180, and KYSE520)?
  6. Figure 2D: showed SNRPB2 staining images. How about CRNKL1 staining images?
  7. Figure 3D: showed SNRPB2 staining images. How about CRNKL1 staining images?
  8. Figure 5J: what is the x axis? days?
  9. Please list all abbreviations.

Reviewer 3 Report

Comments and Suggestions for Authors

Reviewer report for authors of manuscript “Multi-omics analysis of survival-related Splicing Factors and Identifies CRNKL1 as a Therapeutic Target in Esophageal Cancer”

This study investigates the role of survival-related splicing factors (SFs) in esophageal cancer, with a particular focus on CRNKL1 as a key regulator of alternative splicing. The integration of multi-omics data with functional validation provides a strong foundation for the study’s conclusions. However, several aspects require further clarification and additional experimental validation to strengthen the findings. Here are my comments and suggestions:

  • The study identifies six survival-related SFs, but only CRNKL1 is extensively validated. While bioinformatics analyses support the involvement of the other SFs, functional studies (such as knockdown or overexpression) for at least one or two additional SFs would provide stronger evidence for their biological relevance.
  • Have the authors considered investigating whether these SFs work in a coordinated manner, or if their effects are independent?
  • The conclusions regarding CRNKL1’s role in metastasis and cancer progression are based on in vitro assays. In vivo studies, such as xenograft or patient-derived xenograft models, would provide stronger translational relevance. If in vivo experiments are not feasible, the authors should acknowledge this limitation and discuss potential future studies.
  • The study demonstrates that CRNKL1 regulates CTTN and CD44 alternative splicing, but the mechanism remains unclear. Have the authors considered performing RNA immunoprecipitation or CLIP-seq to determine whether CRNKL1 directly binds to these transcripts? Is there evidence that CRNKL1 interacts with the spliceosome machinery in esophageal cancer?
  • In some figures (e.g., tumor microenvironment correlation analyses), the statistical tests and adjustments for multiple comparisons are not clearly detailed. The authors should clarify these aspects to ensure reproducibility.
  • The workflow for alternative splicing event identification should be described more clearly. Were cutoffs applied for differential splicing events?
  • The manuscript is data-heavy, making it difficult to follow in certain sections. Figures could be better labeled to improve clarity, especially those showing pathway enrichment and correlation analyses.
